# A qualitative study exploring how patient engagement activities were sustained or adapted in Canadian healthcare organizations during the COVID-19 pandemic

**Michelle Marcinow**[1,2]*, **Jane Sandercock**[3], **Lauren Cadel**[1,4], **Harprit Singh**[1], **Sara J. T. Guilcher**[2,4], **Penny Dowedoff**[5], **Alies Maybee**[6], **Susan Law**[1,2], **Carol Fancott**[7], **Kerry Kuluski**[1,2]

1 Institute for Better Health, Trillium Health Partners, Mississauga, Ontario, Canada, 2 Institute of Health Policy, Management and Evaluation, University of Toronto, Toronto, Ontario, Canada, 3 Faculty of Rehabilitation Science, McMaster University, Hamilton, Ontario, Canada, 4 Leslie Dan Faculty of Pharmacy, University of Toronto, Toronto, Ontario, Canada, 5 British Columbia's Office of Human Rights Commissioner, Vancouver, British Columbia, Canada, British, 6 Independent Patient Partner, Vancouver, Canada, 7 Patient Engagement & Partnerships, Health Excellence Canada, Ottawa, Ontario, Canada

* michelle.marcinow@thp.ca

## Abstract

### Background

The COVID-19 pandemic caused disruptions across healthcare systems globally exposing the precarious state of patient engagement across all levels of healthcare. While evidence is emerging to describe how engagement was affected across various settings, insights about how some organizations at the policy and practice level of healthcare were able to sustain or adapt patient engagement activities is lacking.

### Objective

This paper addresses the following research question: "*How were healthcare, government, and patient partner organizations able to sustain or adapt patient engagement activities during the COVID-19 pandemic?*"

### Methods

A qualitative descriptive study was conducted to understand how patient engagement activities were maintained or adapted in a variety of healthcare, government, and patient partner organizations in Canada throughout the pandemic. This analysis was part of a larger qualitative, multiple case study where one-to-one interviews were conducted with organizational leaders, managers and patient partners.

### Results

The following themes were identified as key aspects of maintaining or adapting patient engagement activities: 1) having an embedded organizational culture of patient

publicly. Please contact the Trillium Health Partners Research Ethics Board at THPREB@thp.ca to request access to the de-identified data for researchers who meet the criteria for access to confidential data.

**Funding:** This paper was funded by Healthcare Excellence Canada. Dr. Kerry Kuluski holds the Dr. Mathias Gysler Research Chair in Patient and Family-Centered Care at the Institute for Better Health, Trillium Health Partners. The funders had no role in the study design, data collection and analysis, decision to publish, or preparation of the manuscript.

**Competing interests:** The authors have declared that no competing interests exist.

engagement; 2) adapting patient engagement activities to focus on COVID-19 response efforts; 3) having patient partners who exercised leadership and advocacy to support patient care and experiences during the pandemic; and 4) leveraging virtual technology as a communication tool to engage patient partners.

## Conclusion

This paper highlights important insights that may be useful to other health care organizations on how to sustain or adapt patient engagement activities during a healthcare crisis. Having patient engagement embedded within an organization's culture supported by, but not limited to, infrastructure, resources, investments in dedicated staff and patient partner leadership, and communication strategies and tools enabled continued patient engagement activities during the pandemic.

## Introduction

The onset of the COVID-19 pandemic created significant disruptions across healthcare systems globally. While pivoting to address patient care for both patients with and without COVID-19, many organizations rapidly adapted or created new policies and services to decrease the risk of COVID-19 transmission for patients, caregivers, families, healthcare providers and staff, and the broader community [1–5]. Delivering high quality patient care is contingent on engaging patients, caregivers and families to ensure alignment with personal needs and preferences [6], to assure patient-oriented care. Yet, as part of the crisis management during the pandemic, there is evidence of missed opportunities to incorporate these perspectives to support rapid decision-making and the implementation of new processes and policies [7, 8]. Beyond being engaged in their direct care, patients and their caregivers/families can become patient partners in collaboration and partnership with researchers and professionals across all levels of the healthcare system in a range of patient engagement activities, such as planning, research, evaluation, consultation, and decision making [9, 10].

The COVID-19 pandemic affected patient engagement in various ways across all levels of healthcare [8]. In many cases, patient engagement activities, such as patient and family advisory councils, and non-COVID related research and engagement activities that involved patient partners were significantly decreased or halted completely [11–13], and policies and practices were rapidly designed and implemented to support COVID-19 response efforts with limited input from patients [11, 14].

A recent scoping review identified examples of patient engagement during the initial onset of the pandemic, with most examples being at the care delivery level. For example, virtual care consultations grew rapidly given restrictions in face-to-face encounters, with some examples of patient feedback being used to inform the design of virtual care, service or survey development [15]. Some organizations made efforts to keep patients, caregivers and families engaged by holding town halls and forums [16], using virtual forms of engagement [11, 17], or in some cases, involving patient partners in COVID-19 specific research [11, 13]. As well, in a national Canadian survey, patient partners identified both benefits (e.g., technology increased engagement) and challenges (e.g., lack of communication for when activities would recommence) that were experienced in the early stages of the pandemic. While the current literature provides some examples of the impacts of the current COVID-19 pandemic on patient engagement

activities across various healthcare settings [11], what remains unclear is *how* some organizations were able to ensure that patient engagement continued. Understanding the nuances of what worked well or not across organizations at both the policy and practice level of healthcare would be helpful for future planning and sustainability of patient engagement activities during a healthcare crisis. The following qualitative paper is guided by the following research question: "*How were healthcare, government, and patient partner organizations able to sustain or adapt patient engagement activities during the COVID-19 pandemic?*"

## Materials and methods

### Study overview

The study team conducted a qualitative, multiple case study where one-to-one interviews were conducted with stakeholders from a variety of healthcare (e.g., hospital), government (e.g., provincial health authority) and patient partner organizations across Canada (at the policy and practice level of healthcare). The goal of the case study was to obtain a pan-Canadian overview of how patient engagement was impacted by the pandemic within a sample of organizations during COVID-19 (i.e., comparison of patient engagement activities before and during the pandemic, and lessons learned). This paper presents findings from an analysis that focused on understanding what led to patient engagement being sustained during the pandemic. This study was approved by the Trillium Health Partners Research Ethics Board (Study ID#1005).

### Environmental scan

To inform the selection of cases, an environmental scan was conducted and virtual meetings held with key informants in the patient engagement field (including patient partners) who worked in various healthcare, government, and patient partner organizations at the provincial and regional levels [18]. An initial list of key informants was provided by Healthcare Excellence Canada, an independent not-for-profit charity funded by Health Canada, which was simultaneously engaging with organizations and patient partners on the topic of patient engagement during COVID-19. These key informants recommended other individuals and organizations to contact, who were perceived as maintaining patient engagement activities throughout the pandemic. The Principal Investigator (KK) had informal discussions ranging from 30–90 minute phone or Zoom calls with 24 key informants in 21 meetings during May and June 2020. Key informants were located across various provinces, including British Columbia, Alberta, Saskatchewan, Ontario, Quebec and Nova Scotia.

### Case selection and recruitment

Cases were purposely selected based on recommendations made by key informants during the environmental scan. The cases included eight different organizations across five provinces in Canada, including: two provincial health authorities (n = 11 interviews); two provincial healthcare agencies (n = 8 interviews); an acute care academic hospital (n = 3 interviews); a regional integrated care network (n = 4 interviews); a community support agency (n = 4 interviews); a patient partnership network (n = 4 interviews). Three to five people per case was the target recruitment to capture perspectives from both organizational leaders/managers and patient partners [19]. Stakeholders from the environmental scan and the individuals they recommended were contacted for an interview. A snowball recruitment strategy was then used to ask initial interviewees to recommend other stakeholders from their organizations to interview. To be eligible to participate, participants had to be 18 years and older, able to speak and understand English, and willing to participate in a phone or Zoom interview. Individuals could be

involved in patient engagement activities at the local, provincial or national level, including, but not limited to: program planners, patient and family partners, staff in patient relations departments and quality improvement, staff from health organizations (e.g., hospitals) or government agencies.

## Data collection

Most of the participant interviews were conducted between June and October 2020. An additional eighth case was added in March 2021 to broaden geographical representation across Canada. In one case, a patient partner and organizational leader chose to conduct an interview together and a follow-up individual interview was done with each individual to clarify details from the first interview. Participants provided written or verbal informed consent before participating in an interview with a trained qualitative research team member (MM, JS, PD), as per institutional research ethics approval. Consent was documented by the researcher who obtained and documented consent on the consent form and in the recruitment log, both of which were stored separately from the interview. All team members received supervision and mentorship from the Principal Investigator (KK). Interviews were conducted by phone or Zoom and lasted between 30 and 90 minutes in length. An interview guide was developed by the research team to capture information on the participants' role in relation to patient engagement activities, the characteristics of the organization(s) that they worked with, their perception of how patient engagement changed during COVID-19, barriers and facilitators experienced in continuing or adapting patient engagement activities following the onset of COVID-19, as well as overall lessons learned. The interviewers made reflective notes following each interview and discussed each transcript during weekly team meetings to inform subsequent interviews. All interviews were audio-recorded and transcribed verbatim.

## Data analysis

A thematic analysis was used to identify common themes across the stakeholder interviews [20]. Interviews were analyzed inductively, where interviewer statements were used to capture unexpected or emergent themes as well as deductively by identifying the context, processes, barriers, facilitators, and lessons learned related to patient engagement activities during the COVID-19 pandemic. A subset of transcripts was divided amongst the research team (MM, JS, LC, KK, PD) for review with two team members assigned to each transcript to identify core ideas and concepts from the interviews. These concepts were used to inform a preliminary codebook that was used by three team members (MM, JS, LC) to code a subset of transcripts. Minor revisions were made to the codebook after team discussions and the final codebook was applied to the remaining transcripts. In order to explore responses in light of the guiding question, the following four of 17 codes from the codebook were selected for deeper reflection and analysis by the research team (MM, JS, KK, HS, LC): patient engagement activities during COVID-19; how patients were engaged; organizational context; patient engagement structure/ overarching model. Through an iterative process, the research team identified overarching concepts to further categorize the codes. A series of broader team discussions and reflections further organized these concepts to form the themes discussed in this paper.

## Results

Thirty-four participants were interviewed across 8 case sites located across five Canadian provinces (British Columbia, Saskatchewan, Ontario, Quebec and Nova Scotia). Participants held a variety of different roles within these organizations: eight organizational leaders (i.e., such as director or vice president), nine managers (i.e., other types of leaders with portfolios that

Table 1. Themes and sub-themes.

| Themes | Sub-themes |
|---|---|
| Having an embedded organizational culture of patient engagement | Patient engagement was an organizational priority prior to the pandemic |
| | Embedded frameworks and structures helped to support engagement |
| Adapting patient engagement activities to focus on COVID-19 response efforts | Leaders informally engaged with patient partners for feedback on COVID-19 decisions |
| | Patient partners were invited to join COVID-19 response panels and committees |
| Having patient partners who exercised leadership and advocacy to support patient care and experiences during the pandemic | Grassroots efforts led by patient partners helped continue engagement |
| | Patient partners raised concerns about pandemic related decision-making |
| Leveraging virtual technology as a communication tool to engage patient partners | Virtual technology was implemented to maintain momentum with engagement |
| | Virtual engagement increased accessibility for patient engagement |
| | Virtual technology had its drawbacks due to learning curves |

included some aspect of patient engagement, including strategic or program leads and advisors), and 17 patient partners. Two of the participants had dual roles within their respective organizations acting in a leadership or manager type role in addition to being a patient partner. Across organizations, it was noted that there were different terms used by interviewees to identify patients, caregivers and family members who worked in partnership with researchers or health care professionals (e.g., patient and family advisors); however, the term patient partner will be used for consistency throughout the manuscript when referring to patient and caregiver partner interviewees. Described below are four key themes (Table 1) that were identified through the analysis that helped us understand how healthcare organizations were able to sustain or adapt patient engagement activities during the pandemic: 1) having an embedded organizational culture of patient engagement; 2) adapting patient engagement activities to focus on COVID-19 response efforts; 3) having patient partners who exercised leadership and advocacy to support patient care and experiences during the pandemic; and, 4) leveraging virtual technology as a communication tool to engage patient partners. Each main theme (Table 1) is reported below with accompanying quotes to substantiate the findings [21].

## Having an embedded organizational culture of patient engagement

Participants described their affiliated organizations as having a culture that is committed to patient engagement, with many participants providing historical context about the evolution of patient engagement and how it came to be a fundamental practice within their organization. The tangible elements that were identified as having an embedded culture included having leadership support for engagement, structures and resources being put towards, and remaining available for engagement, and having a guiding framework or approach at the organization that created a unified approach and commitment to engagement.

**Patient engagement was an organizational priority prior to the pandemic.** Many of the organizations had working groups or teams made up of leaders, staff, and patient partners who advocated for their organizations on the importance of patient engagement and supported patient engagement activities. One participant reflected on their organization's philosophy of patient centeredness and how that translated into engagement and partnership:

*"Patient/family centred care is actually our philosophy of care. So, we have–our mission vision values, we have all–you know, of those kind of reflect patient/family centred care but the [provincial health authority] actually calls [it] out [in] our philosophy–kind of like a little add-on piece–where we say that our philosophy of care is patients and families centred.*

They further went on to say: *"My staff are the ones that [are] leading engagement work on the frontline, recruiting patient/family advisors, working with leadership to include patients, families and all of their processes for program design, process design, any kind of redesign making sure that we have patients and families involved."* (Participant #27, organizational leader, provincial health authority).

Participants also described leadership support of the organizational commitment to engage with patient partners. One leader described their outlook on patient engagement:

*"This work is really about an extension of how we live our mission within the organisation, so it's all about why we do what we do [. . .] It's not just about doing the right thing for the organisation, we live in the community, and we seek care and within our healthcare system, as do our loved ones. So, it's in everyone's interest. I realise it sounds a bit apple pie motherhood, but I–that's how I feel"*

*(Participant #3, organizational leader, acute care academic hospital)*

**Embedded frameworks and structures helped to support engagement.** Having both frameworks and structures in place and leadership support was in some cases described by participants as fundamental to sustaining or adapting patient engagement activities during the pandemic. For example, one organization had a formal paid role of a patient and family centered care lead who liaised between patient partners and the leadership team. This lead continued to act as a liaison throughout the pandemic and was not redeployed, an important structure in place that enabled continued engagement. This participant reflected on the reasons that patient partners felt ready to support the organization during the healthcare crisis:

*"We suddenly found ourselves in this situation of decision making happening on the fly, really fast, with really tight timelines, which changed how we had to engage. And there was, because of the existing foundation of engagement in the organisation, there was a trust that actually brought out in the advisors, a desire to help and provide feedback quickly and really thoughtful feedback to the organisation that really had, I think fairly–I think ongoing and ongoing input and influence in the decisions that were being made."*

*(Participant #1, patient partner, acute care academic hospital)*

Another participant described a framework within their organization that supports the commitment to patient engagement even during a crisis:

*"It is acknowledgement that, yep, we need to focus on the crisis and then we need to switch in to make sure we are engaging with patients to understand [bucket] one, how is this crisis affected? Bucket two. How are we communicating to them about the crisis? Bucket three, we have to set up a council to make sure we are acknowledging the impact of the crisis, et cetera. So that's more of the framework I'm referring to, is specifically, you know, when something sways an organization from their normal mode of operation, how does patient engagement work?"*

*(Participant #16, patient partner, provincial health agency)*

## Adapting patient engagement activities to focus on COVID-19 response efforts

The growing crisis created a sense of urgency among organizations to react in a timely manner to support the pandemic response. In some cases, as a response to the healthcare crisis, patient partners were informally engaged by leaders to provide feedback on COVID-19 response efforts. Patients, caregivers and families were often directly impacted by COVID related decisions and thus provided important insight as members of response panels and committees to support decision-making during the pandemic.

**Leaders informally engaged with patient partners for feedback on COVID-19 decisions.** Those in leadership positions helped pivot engagement activities to support COVID-19 response efforts. For example, prior to widespread COVID-19 lockdowns, one leader reached out to patient partners within their organization and met off-site to learn about what priorities were important to patient partners for the organization to focus on while anticipating upcoming mandates and restrictions:

> *"[The leader] pulled together a group of patient experience advisors and [had] us meet to give her kind of our reactions to issues that she suspected that she might be heading into as the pandemic developed and that was the first time that I can think of, that a senior executive in the hospital, being proactive in terms of reaching out beforehand and not just reaching out for information, but obviously to create some relationships in the network."*

> *(Participant #2, patient partner, acute care academic hospital)*

At another organization, which had initially implemented a no-visitor policy based on provincial public health guidelines, leaders engaged directly with families and caregivers at entry screening desks to understand how their no-visitor's policy had restricted caregiver supports for their clients (e.g., obtaining groceries and medication). They used this understanding to modify the no-visitor's policy and allow for caregivers to be considered essential care partners:

> *"Basically the discussion we had was if we can redeploy people who are volunteers into caregiving roles for our clients, why can't we allow caregivers who are caregivers and who care about these people and don't want to get them sick, they're a loved one, they're a friend, they have a vested interest in not bringing Covid into our community you know, and they already know this person. Like how can we not come up with something to allow these folks to come in? So that's where we saw the opportunity was let's implement what we called our Caregiver Presence policy."*

> *(Participant #7, organizational leader, community support agency)*

Some patient partners were also approached by leadership to lead communication efforts within the community (e.g., hosting webinars) and develop resources to help the general public cope with the pandemic and navigate the healthcare system:

> *"The first webinar that we held, the topic was around supporting caregivers during the pandemic and what it kind of means to them. And so, they [patient partners] designed the agenda, we had actually one of our caregiver partners who's kind of a caregiver extraordinaire, [. . .], he actually was one of the featured speakers on that and he was talking about technology enablers to support caregiving."*

> *(Participant #4, manager, regional integrated care network role)*

**Patient partners were invited to join COVID-19 response panels and committees.**
While many pre-pandemic patient engagement activities were paused, some patient partners
were invited by organizational leaders to join executive panels or COVID-19 committees to
provide feedback on decisions being made during the pandemic:

*"One of the first activities that we did back in April, was to review some documents from the
[Ministry of Health] on the potential lack of resources if the [COVID-19] cases became so
large that we wouldn't have enough equipment or human recourse to take care of every-
body. What do you, you know who do you take care of, how do you prioritize your cases.
Already we saw the panic in Europe, and we were looking at the prioritization that cur-
rently exists in the protocols here in Quebec and we were asked to review that with patients'
eyes."*

*(Participant #13, patient partner, patient partnership network)*

On some committees, patient partners were involved in co-designing tools or guidance
documents and providing support for staff while navigating newly implemented policies:

*"We had, because we have four patient and family advisors on our expert panel that was
meeting weekly, it's now meeting every second week, but those patient and family advisors
have been co-designing the policy, all of the resources, the strategies and any supporting mate-
rial [e.g., strategic document outlining key areas of focus of the patient experience portfolio].
We also formed the family presence support team, so that support team is responsible for a
consultation service, so if there is providers that have questions about family presence and the
current policy, they call the support team for guidance. And the support team also facilitates
an appeal process for patients and families, so you know, having a patient and family advisor
on that group has been really important."*

*(Participant #22, organizational leader, provincial health authority)*

## Having patient partners exercise leadership and advocacy to support patient care and experiences during the pandemic

Patient engagement activities in some organizations were able to continue because patient
partners demonstrated leadership by leveraging their networks to push forward engage-
ment ideas or quickly step in to support the COVID-19 response. Patient partners some-
times found themselves in a position where they either had to take the lead in pushing
forward patient engagement activities or speak up about decisions being made that could
negatively impact the care and experiences of patients, caregivers and families during the
pandemic.

**Grassroots efforts led by patient partners helped continue engagement.** When patient
engagement activities were halted in one organization, there were grassroots efforts made by
patient partners who began working together to mobilize their own patient partner networks
to drive forward patient engagement activities themselves. One patient partner described how
their network leveraged virtual technology and designed and launched interactive webinars
educating how patient partners and staff could work together (e.g., how to build meaningful
connections and relationships). The webinars attracted 120 attendees from across Canada
showcasing the leadership of this patient partner community and how they were able to create
meaningful resources that had a far reach across the nation:

*"We came up with the idea of starting a webinar series. And we've had two of them and the last one we just had a few weeks ago and it's been an unbelievable success. Like the last one, I mean we opened it up to people across Canada and its staff and patient family advisors."*

*(Participant #31, patient partner, provincial health authority)*

In another example, a patient partner was called on by healthcare leaders to support the COVID-19 response. This patient partner built a strong foundation of patient engagement in their province along with trusted relationships with the government and hospital leaders. The patient partner described one instance where they were asked by health care leaders to be on an ethics committee and focused on navigating difficult ethical topics that arose during the pandemic, which also included further engaging other patient partners on the matter:

*"There was a lot of work driven by this committee and those patients were engaged. And actually, we constituted a patient expert group, you know, specifically for COVID and we selected the patients who had an experience related to the crisis."*

*(Participant #8, organizational leader & patient partner, patient partnership network)*

**Patient partners raised concerns about pandemic related decision-making.** Patient partners spoke up and advocated on behalf of other patients and their caregivers and families when they observed decisions during the pandemic that may be considered hurtful. For example, one patient partner noticed signs were posted at the visitor screening station of the hospital that advised visitors to have respect and dignity for hospital staff. Family members perceived this signage to be hurtful because they were trying to share with hospital staff how visitor restrictions were negatively impacting their ability to provide essential care to their family member in hospital. The patient partner felt that the organization took a step backward after previously focusing efforts on removing inappropriate signage in hospital:

*"We have worked so hard over the years to take out negative signs, and here they've put up one that is the first pillar of patient family centred care, respect and dignity. And now using it against patients and families at their worst possible time, during a pandemic. So, I raised this concern several times, and at first–I hate to say it, but I was told to be patient, and we'll work with the staff, just be patient. And finally, I wasn't patient anymore."*

*(Participant #28, patient partner, provincial health authority)*

This led to a conversation between the patient partner and the organization's CEO, who agreed that the signs needed to come down:

*"It was a Monday I spoke to him, and we both agreed that was not appropriate. And he said, if you come for treatment Wednesday, if they're still there please let me know. . .And so when I–when I went on Wednesday they were still up. So, I sent a message to his admin, and I said, just let [CEO] know that they're still up there, and she said, oh my goodness, this is not good. So, she sent a letter to him, and also to the VP of infrastructure. And when I left three hours later, they were down."*

*(Participant #28, patient partner, provincial health authority)*

## Leveraging virtual technology as a communication tool to engage patient partners

Given that orders were implemented by provinces across Canada to reduce transmission by keeping people at home during the early stages of the pandemic, most organizations switched to virtual patient engagement using platforms such as Zoom or Microsoft Teams. The use of virtual technology had its benefits and challenges. In some cases, it helped create an avenue for engagement to continue when in-person was no longer possible, while also creating more flexibility to attend meetings for some patient partners. On the other hand, switching to virtual engagement was a steep learning curve for both organizations and patient partners.

**Virtual technology was implemented to maintain momentum with engagement.** Virtual meetings were often held in place of in-person meetings in an attempt to sustain momentum with patient engagement or simply as a wellness check with patient partners. Organizations that successfully adapted patient engagement activities during the pandemic took time to consider how to modify patient engagement activities to an online format:

> *"We all looked at every program and said what do we need to do now? Which ones are fine? Which ones we need to change. And, as I said, just saying that oh, we're going to take this program and move it online doesn't mean you just, like PDF some materials and putting them online. It's how do you actually make it effective? So, we were working with all of our partners, in communication with partners, just to go through everything to say what's going to be online? How will we do it? How will we engage people? What new tools and resources do we need this year?"*

> *(Participant #14, manager, provincial health agency)*

The use of virtual technology helped organizations obtain rapid feedback from patient partners to support fast decision-making for implementing new policies, procedures, supports and/or resources. This rapid exchange of information was meaningful for patient partners because they could see how their feedback was being used for quick action. Participants also described how it was fairly easy to jump on a virtual call or to mobilize a group of people in a virtual meeting to discuss initiatives rather than waiting for the next scheduled meeting. One patient partner described how patient partner feedback was obtained quickly by the Ministry of Health to support the launch of a new health application during the pandemic:

> *"And they got us all [on a] call within, like, a day, and they needed our feedback on the app, like, literally in, you know, 24, 48 hours. And then they needed us to view it again before it went live. I mean, this is all within the process of, like, I don't know, four, five days. And there was such feedback from it from patient partners because all they wanted to do was help and get involved. And I think it showed that, you know, when you have an ask, don't think time is a barrier to patient engagement."*

> *(Participant #16, patient partner, provincial health agency)*

**Virtual engagement increased accessibility for patient engagement.** Virtual engagement allowed many patient partners opportunities to participate. Many participants discussed that the limitations to attending meetings at a physical location, such as commuting, balancing personal time commitments, feeling unwell due to illness or having a disability, were non-existent because patient partners could be engaged no matter where they were located. With no

limitations to in-person attendance, many patient partners were able to meet with other patient partners, leaders and staff who they normally would not have interacted with before the pandemic. As a result, virtual engagement created many new relationships and opportunities for co-design and collaboration:

> *"They [patient partners] became I think closer with each other and with us as well in terms of just from a relationship-building perspective. They were able to support each other peer to peer even, able to be a little bit of a conduit in terms of they're each part of their own sort of social and professional and other networks and they were able to almost do some network-to-network networking [. . .]. And then the group became kind of an ad hoc and now becoming a more formal almost place for other partners to bring their initiatives for co-design input."*
>
> *(Participant #4, manager, regional integrated care network)*

**Virtual technology had its drawbacks due to learning curves.** Despite some positive aspects to virtual engagement, participants acknowledged that there were many disadvantages. Virtual technology was implemented quite quickly across many organizations, which created a learning curve for many. For some, lack of access to computers, tablets, reliable and/or affordable internet, and/or a private space at home also created limitations. Some leaders discussed how they were aware of these limitations and made efforts to use different forms of communication, such as phone calls, emails, mail, and newsletters, to keep patient partners engaged and informed. In one organization, financial support for patient partners was provided:

> *"Sometimes we purchase phone cards for people to have data. Like we do–that's kind of our workaround. Like some people have cellphones. It's more the data is the issue. So, we do things like that. We support people that way."*
>
> *(Participant # 11, manager, regional integrated care network)*

## Discussion

This paper provides insights about how patient engagement activities were sustained during the COVID-19 pandemic in eight provincial or regional healthcare organizations across five Canadian provinces. Thirty-four participants were interviewed who were patient or caregiver partners or individuals in leadership or management positions at these organizations to understand what key aspects were considered essential to engaging patient partners during the healthcare crisis. Continuing engagement activities were contingent on: 1) having an embedded organizational culture of patient engagement; 2) adapting patient engagement activities to focus on COVID-19 response efforts; 3) having patient partners exercise leadership and advocacy to support patient care and experiences during the pandemic; and, 4) leveraging virtual technology as a communication tool to engage patient partners. Our observations highlight the on-going work and commitment required by organizations to ensure that patient engagement is not an afterthought during a disruption to the healthcare system.

Having embedded structures in place (e.g., patient engagement frameworks, paid patient partner leadership roles) and leadership commitment to patient engagement were key factors in supporting continued patient partner involvement. Including patients in organizational engagement activities despite pandemic disruptions suggested that patient engagement was valued and prioritized by the organization. Other researchers have discussed that when patient engagement activities were abruptly discontinued during the pandemic, even if embedded

within the organization, this sent a signal to patient partners that their involvement was not needed and patient engagement was not an organizational priority [11, 22]. Organizations will need to consider what structures and resources (e.g., infrastructure to support their engagement practices) are necessary to support on-going patient engagement [9, 23–25]. For example, individuals in leadership roles can support their organization's commitment to patient engagement across all levels of the organization [23, 25] by being a leader who spends the time building collaborating with patient partners to develop adaptive solutions to overcome challenges within the organization to advance patient-centered care [26, 27].

Organizations also have a responsibility to partner with their patients, staff, and community, especially through a healthcare crisis. In doing so, organizations should be proactive and reflect on how they can engage with patient partners to inform crisis management and recovery [28–30]. Patient partners have expertise that is rooted in their interactions with the healthcare system and can provide helpful knowledge about how patients and their caregivers and families are being affected by changes in services and policies as a result of a disruption [6]. Participants reported that while some patient engagement activities were paused, patient partner involvement was re-directed in other ways to support COVID-19 response efforts (e.g., joining COVID-19 working groups, providing input into policy changes, and creating COVID-19 information webinars for the community). This paper showed that patient engagement was a vital and instrumental resource to help navigate the healthcare system during the COVID-19 pandemic. Similarly, a commentary by Richards and Scowcroft (2020) discussed how patient partners can provide insights from the broader community including how the pandemic affects patients and their caregivers and families across various determinants of health to inform how to prioritize and restructure care services during restrictions and lockdowns [7]. Not including patient partners' experience and expertise in response efforts to a healthcare crisis is a missed opportunity. For example, Tripp et al. (2022) reported that some patient partners identified in a Canadian national survey that they were feeling frustrated that they were left out of COVID-19 decision-making and policy development, particularly at a time when decisions had a significant impact on patient experience and care [11]. In other situations, patient partners were heavily relied on to support COVID-19 efforts, and these individuals reported that they were able to see the impacts of their contributions to the organization because of how quickly things would change throughout the pandemic [11].

While the findings of this paper highlight many examples where engagement continued or was adapted, examples where patient partners took initiative to stay engaged and keep momentum were also noted. In some situations, patient partners did not wait to be called on to support COVID-19 responses and were proactive in identifying to leadership when new policies were negatively affecting patient care and experience and offered ideas to revise policies to be more patient-centered. In other examples when engagement was minimal or non-existent, there were examples where patient partners took initiative to develop and launch tailored educational online webinars for various audiences (e.g., patients and families in the local community, patient partners and organizational staff). These examples showcase that leadership within a healthcare organization also comes from patient partners, particularly when it comes to decisions that affect how patients experience health care. There needs to be a change in mindsets where patients are considered as decision-makers and collaborative partners instead of passive consumers of the healthcare system [6]. For example, within and across organizations in England's National Health Service, McNally et al. (2015), observed that patient partners who work in leadership roles alongside clinical and healthcare leaders can have real impact on how patient feedback is used by organizations to improve patient care and experience [26]. As with any patient role created at an organization that is trying to embed meaningful patient engagement [31], patient partner leadership roles should have

organizational support. For example, organizations could invest in the training and development of patient partner leaders and create paid, well-defined formalized leadership positions for patient partners that help create conditions for change from the inside. An organizational culture that respects and incorporates the voices of different type of stakeholders will enhance their capability for collective learning and potentially make greater strides in improving the quality and safety of the healthcare services provided [32].

Not surprisingly, the rapid transition to the use of virtual technology to engage with patient partners during the COVID-19 pandemic was frequently discussed by participants. Many participants highlighted the benefits of using virtual platforms for engagement, including but not limited to, eliminating barriers to in-person engagement (e.g., being able to connect from anywhere), creating new connections and collaborations within and across organizations both locally and nationally, and encouraging organizations to invest in online communication tools and platforms. These benefits echo similar findings presented in the literature [8, 11, 17]; however, one advantage that stood out across the interviews was that virtual engagement created opportunities to quickly mobilize patient partners for feedback related to rapid COVID-19 decision-making and allow patient partners to see how their feedback was being used in real time. Leveraging virtual technology to engage with patient partners can help with 'closing the loop' where patient partners are acknowledged for their time and contributions, are updated on progress being made, told if outcomes were met, and how their feedback has influenced decision-making [33, 34]. Closing the loop is an important component of patient engagement that shows respect and appreciation for patient partners and also allows for health care partners to reflect on what went well or not while holding themselves accountable to their commitment to patient engagement [33].

Although maintaining the patient voice throughout a healthcare crisis is possible by leveraging virtual technology [17, 24], it will not always be seamless as reflected in the results of this paper and barriers reported in the literature (mainly access issues) [11, 15]. A variety of communication methods should ideally be used to engage with patient partners. For example, phone calls, emails, or in person check-ins (if safe to do so) are other ways to engage with patient partners who many not be able to connect through virtual calls. More broadly, organizations can hold virtual town halls or email/mail out newsletters to keep the patient community connected and informed. Technological support and resources should be made available by the organization to ensure equitable access for patient partners to be engaged [11] and adhere to accessibility standards based on different sensory capabilities and learning needs of partners. As well, individuals in the organizations included in this study may have had preexisting relationships which contributed to a smoother transition to virtual platforms and modes of working. Organizations will need to consider how to build relationships and trust with patient partners via virtual means to ensure that they feel comfortable participating in patient engagement activities and get to know other patient partners and organizational leaders and staff as they come to work together.

## Strengths and limitations

To our knowledge, there has been little information published to date about what happened to patient engagement during the pandemic. The findings of this paper shed light on how patient engagement was affected when healthcare organizations were forced to focus on the crisis at hand. This paper highlights what factors helped support patient engagement to continue and how patient partners could be engaged to support COVID-19 efforts when patient engagement activities had to pause. These findings provide useful learnings for other organizations. However, the study sample for this study was drawn from an initial selection of those perceived to

be leaders in patient engagement who sustained engagement in some form during the pandemic. Organizations that experienced significant challenges in maintaining patient engagement activities were not sought out as part of this study, which would have provided additional learnings on how to support engagement during a healthcare crisis. The study sample also did not include long-term or primary care organizations which may have highlighted challenges and learnings that are unique to each sector and/or additional learnings that could be applied across the healthcare system. Furthermore, the majority of the interviews were conducted in between the first and second COVID-19 waves in Canada and it would be helpful for future studies to learn how patient engagement activities evolved or were adapted throughout later waves of the COVID-19 pandemic. The patient partners that were interviewed identified that there was a lack of diverse patient partners within their organization with respect to various attributes, such as socioeconomic status or cultural background, but were working towards diversifying these perspectives within their respective organizations. Thus, the patient partner sample may not have been representative of the range of experiences that patient partners may face when engaging with healthcare organizations as a partner, including those from equity deserving groups, most impacted by the COVID-19 pandemic itself. Lastly, interviews were done by Zoom or telephone and may have limited the interviewer from observing nonverbal cues; however, the literature suggests that using both of these methods within the same study does not undermine the quality of the methodology [35–37].

## Conclusion

This paper explored how patient engagement was impacted across a sample of Canadian organizations that focus on healthcare at a policy or delivery level to better understand what enabled these organizations to continue patient engagement. The following key takeaways were identified: 1) Organizations should require not only having frameworks and resources in place but also leadership commitment to supporting patient engagement.; 2) Patient partners are a vital resource who engaged in COVID-19 response efforts to help guide organizations through a healthcare crisis, particularly regarding decisions that directly impacted patients and their families; 3) Organizations can nurture and support the development of leadership qualities of patient partners who can be strong advocates for maintaining quality patient care and experience during health crises; and, 4) Organizations should leverage virtual technology to keep the lines of communication open with patient partners to support continued engagement when in person engagement is not possible. Keeping in mind however that technology can also present challenges that may limit engagement for some patient partners. Having open communication with patient partners is suggested to learn how best to engage during a time of disruption.

The findings from this paper highlight that by engaging patient partners during a health crisis, impacts can be seen in real time both by the organizations and the patient partners themselves. Engaging patient partners is not something that can be implemented overnight, but with each organization having a history of engagement and organizational cultures that valued partnership likely made adaptability of engagement activities easier. Other organizations will need to commit their time to creating a culture that values patient engagement, but doing so will help organizations better support patient engagement activities regardless of the state of the healthcare system.

## Acknowledgments

The authors would like to thank Aditi Desai for her support on this project. We would also like to thank all the participants for generously sharing their time, experiences and knowledge to support this work.

## Author Contributions

**Conceptualization:** Michelle Marcinow, Jane Sandercock, Kerry Kuluski.

**Data curation:** Michelle Marcinow, Jane Sandercock, Lauren Cadel, Penny Dowedoff, Kerry Kuluski.

**Formal analysis:** Michelle Marcinow, Jane Sandercock, Lauren Cadel, Harprit Singh, Penny Dowedoff, Kerry Kuluski.

**Funding acquisition:** Carol Fancott, Kerry Kuluski.

**Investigation:** Michelle Marcinow, Jane Sandercock, Penny Dowedoff, Kerry Kuluski.

**Methodology:** Kerry Kuluski.

**Project administration:** Michelle Marcinow, Jane Sandercock, Penny Dowedoff, Kerry Kuluski.

**Supervision:** Kerry Kuluski.

**Validation:** Sara J. T. Guilcher, Kerry Kuluski.

**Visualization:** Michelle Marcinow, Kerry Kuluski.

**Writing – original draft:** Michelle Marcinow, Jane Sandercock, Lauren Cadel, Harprit Singh, Sara J. T. Guilcher, Kerry Kuluski.

**Writing – review & editing:** Michelle Marcinow, Jane Sandercock, Lauren Cadel, Harprit Singh, Sara J. T. Guilcher, Penny Dowedoff, Alies Maybee, Susan Law, Carol Fancott, Kerry Kuluski.

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
