## [Decision Letter · Decision Letter 0]

22 Dec 2022

PONE-D-22-27751A qualitative study exploring how patient engagement activities were sustained or adapted in Canadian healthcare organizations during the COVID-19 pandemicPLOS ONE

Dear Dr. Marcinow,

Thank you for submitting your manuscript to PLOS ONE. After careful consideration, we feel that it has merit but does not fully meet PLOS ONE’s publication criteria as it currently stands. Therefore, we invite you to submit a revised version of the manuscript that addresses the points raised during the review process.

The reviewers have highlighted strengths in this manuscript, but one reviewers still has major comments that need to be addressed before I can make final decision on this publication. Thanks for taking them into consideration and address them in detail. 

We look forward to receiving your revised manuscript.

Kind regards,

Sara Rubinelli

Academic Editor

PLOS ONE

Journal Requirements:

When submitting your revision, we need you to address these additional requirements. 1. Please ensure that your manuscript meets PLOS ONE's style requirements, including those for file naming. The PLOS ONE style templates can be found at https://journals.plos.org/plosone/s/file?id=wjVg/PLOSOne_formatting_sample_main_body.pdf and https://journals.plos.org/plosone/s/file?id=ba62/PLOSOne_formatting_sample_title_authors_affiliations.pdf  2. Thank you for stating the following financial disclosure: "This paper was funded by Healthcare Excellence Canada. Dr. Kerry Kuluski holds the Dr. Mathias Gysler Research Chair in Patient and Family Centred Care at theInstitute for Better Health, Trillium Health Partners. The funding for this Chair, supported by the Trillium Health Partners Foundation, was used to support Dr. Kuluski’s time in leading the study reported in this paper." 
 Please state what role the funders took in the study.  If the funders had no role, please state: "The funders had no role in study design, data collection and analysis, decision to publish, or preparation of the manuscript." If this statement is not correct you must amend it as needed. Please include this amended Role of Funder statement in your cover letter; we will change the online submission form on your behalf.  3. Please note that in order to use the direct billing option the corresponding author must be affiliated with the chosen institute. Please either amend your manuscript to change the affiliation or corresponding author, or email us at plosone@plos.org with a request to remove this option.

Reviewers' comments:

Reviewer's Responses to Questions

**Comments to the Author**

1. Is the manuscript technically sound, and do the data support the conclusions?

Reviewer #1: Yes

Reviewer #2: Partly

2. Has the statistical analysis been performed appropriately and rigorously? 

Reviewer #1: N/A

Reviewer #2: N/A

3. Have the authors made all data underlying the findings in their manuscript fully available?

Reviewer #1: Yes

Reviewer #2: No

4. Is the manuscript presented in an intelligible fashion and written in standard English?

Reviewer #1: Yes

Reviewer #2: Yes

5. Review Comments to the Author

Reviewer #1: This paper is a quality job, and the results are very useful. I have a little to add about qualitative methodology.

It would be interesting for authors and readers if authors compare their conclusions with relevant quantitative studies and qualitative. In this way, they can assess and value if conclusions can be helpful for quantitative in the future.

Authors could argue more if these conclusions and comments of their work can help to create sustainable healthcare organisations in the future, and these can lead to more innovative procedures.

I suggest you look at the following papers

https://doi.org/10.3390/su142215416

https://www.mdpi.com/2071-1050/12/7/2730

Reviewer #2: Dealing with the pandemic ad still involving the patients of different groups is a topic of great interest. The authors try to address this topic with a qualittive approach conducting qualitative study phases. This is interesting, even if only for a small target group.

The manuscript is well-written but needs major revisions. For me there are a few questions that need to be addressed.

General:

Spacing and orthographies: Throughout the manuscript there are various spacing issues.

Precision: Throughout the manuscripts various paragraphs are very talkative and too long. More actual information would be beneficial. Please be more precise and give necessary information.

Abstract

p.2, l. 32: What is defined as “organizations” in this case?

Introduction:

The manuscript lacks some links between the introduction section and its theories with the final results of the current study. This should be more precise.

Throughout the sections “introduction”, “methods” and “results” there are several occasions where authors use personal statements (we, our, etc) This should be deleted. Personal statements and comments should only be included in the discussion section.

Materials and Methods:

p. 4, ll.83: This has been said a couple of time before. The authors should focus on more precision and avoid giving certain statements at different pages of the manuscript again and again.

p. 5, ll.94: “various healthcare organization”. What is meant by this? What define the authors as healthcare organization? So far, the reader have no idea on where the interviews were conducted. It’s a big difference if local support groups or a university hospital manager was interviewed.

p. 5, ll.94: Why informal discussions? That is not scientifically sound. Was an interview guide used? Where the discussions transcribed?

p. 6, l. 109: “eight different organizations”. Please clarify the process. Which organizations?

p. 6, l.114ff: Concerning the eligibility criteria: What about consenting? Was consent obtained? Where participants included without consent?

p. 6, l. 125: verbal consent: See comment above. A verbal consent is useless. Respondents need to give written consent in order to use the data.

p. 7, l. 146ff: There seems to be a problem with formatting. The color is changing.

Results:

In general: This section is rather challenging and needs more explanation

p. 8, l. 165: For this study in general: The manuscript would highly benefit from a clear definition what authors mean by “partnership” and “patient activity”. This should be added

The whole section is way too long and uses way too many quotes. There are no actual results presented.

Discussion:

In general, the discussion of the results can be shortened. There are several paragraphs that just repeat statements given I the results section already. The authors should be more precise and discuss the impact of there findings. What is the take home message? What can other hospitals learn from the examples presented.

p.25, l.537: This is already mentioned on page 20, lines 412ff..should be deleted here

p. 26, l. 552-554: This has been stated several times before. Please be more precise and avoid repetitions.

Conclusion:

p. 27, l. 575: please see comment on p. 26, l. 552-554. This is said again and again. -

The conclusion is too vague and too generally valid and not specifically adjusted to the study. More details on how the study finding add value to the ongoing discussion should be given.

At the beginning of the manuscript a multi-center study is mentioned. How do the results of this study part influence to overall study?

References:

Reference 33: Why is the PMID given?

6. PLOS authors have the option to publish the peer review history of their article (what does this mean?). If published, this will include your full peer review and any attached files.

Reviewer #1: No

Reviewer #2: No

---

## [Author Response · Author response to Decision Letter 0]

27 Jan 2023

We are thankful to the reviewers for their time and consideration of our manuscript. The reviewer feedback helped to strengthen the original manuscript for re-submission. We have included a Response to Reviews attached with this re-submission that responds to each reviewer question or comment.

---

## [Decision Letter · Decision Letter 1]

27 Feb 2023

A qualitative study exploring how patient engagement activities were sustained or adapted in Canadian healthcare organizations during the COVID-19 pandemic

PONE-D-22-27751R1

Dear Dr. Marcinow,

We’re pleased to inform you that your manuscript has been judged scientifically suitable for publication and will be formally accepted for publication once it meets all outstanding technical requirements.

Kind regards,

Sara Rubinelli

Academic Editor

PLOS ONE

Additional Editor Comments (optional):

Reviewers' comments:

Reviewer's Responses to Questions

**Comments to the Author**

1. If the authors have adequately addressed your comments raised in a previous round of review and you feel that this manuscript is now acceptable for publication, you may indicate that here to bypass the “Comments to the Author” section, enter your conflict of interest statement in the “Confidential to Editor” section, and submit your "Accept" recommendation.

Reviewer #1: All comments have been addressed

Reviewer #2: All comments have been addressed

2. Is the manuscript technically sound, and do the data support the conclusions?

Reviewer #1: Yes

Reviewer #2: Partly

3. Has the statistical analysis been performed appropriately and rigorously? 

Reviewer #1: N/A

Reviewer #2: N/A

4. Have the authors made all data underlying the findings in their manuscript fully available?

Reviewer #1: No

Reviewer #2: Yes

5. Is the manuscript presented in an intelligible fashion and written in standard English?

Reviewer #1: Yes

Reviewer #2: Yes

6. Review Comments to the Author

Reviewer #1: The authors have addressed all the main comments, and many issues have been clarified enough in the revised version.

Reviewer #2: The authors have adequately addressed the comments raised in the previous round of review and this manuscript is now acceptable for publication

7. PLOS authors have the option to publish the peer review history of their article (what does this mean?). If published, this will include your full peer review and any attached files.

Reviewer #1: No

Reviewer #2: No

---

## [Editor Report · Acceptance letter]

7 Mar 2023

PONE-D-22-27751R1 

A qualitative study exploring how patient engagement activities were sustained or adapted in Canadian healthcare organizations during the COVID-19 pandemic 

Dear Dr. Marcinow:

I'm pleased to inform you that your manuscript has been deemed suitable for publication in PLOS ONE. Congratulations! Your manuscript is now with our production department. 

Kind regards, 

on behalf of

Dr. Sara Rubinelli 

Academic Editor

PLOS ONE